# Evaluating Advanced Large Language Models for Pulmonary Disease Diagnosis Using Portable Spirometer Data: A Comparative Analysis of Gemini 1.5 Pro, GPT 4o, and Claude 3.5 Sonnet

Jin-Hyun Park
*Department of Biomedical Informatics*
*Korea University College of Medicine*
Seoul, Republic of Korea
pjhjin43@korea.ac.kr

Chinock Cheong
*Division of Colon and Rectal Surgery*
*Korea University College of Medicine*
Seoul, Republic of Korea
owho9@naver.com

Sanghee Kang
*Division of Colon and Rectal Surgery*
*Korea University College of Medicine*
Seoul, Republic of Korea
kasaha1@korea.ac.kr

Inpyo Lee
*BREATHIGS Co., Ltd.*
Wonju, Republic of Korea
ceo@breathings.co.kr

Sungjin Lee
*BREATHIGS Co., Ltd.*
Wonju, Republic of Korea
gordon@breathings.co.kr

Kisang Yoon
*BREATHIGS Co., Ltd.*
Wonju, Republic of Korea
evan@breathings.co.kr

Hwamin Lee
*Department of Biomedical Informatics*
*Korea University College of Medicine*
Seoul, Republic of Korea
hwamin@korea.ac.kr

*Abstract*— **Pulmonary function tests (PFTs) are vital for diagnosing various pulmonary conditions, including chronic obstructive pulmonary disease (COPD) and asthma. Traditional PFTs, conducted using laboratory-based spirometers, are accurate but costly and require skilled technicians. Recent advancements in portable spirometry and large language models (LLMs) offer promising alternatives for remote diagnostics and clinical decision support. This study evaluates the performance of three advanced LLMs: Gemini 1.5 Pro, GPT 4o, and Claude 3.5 Sonnet in understanding and interpreting PFTs data. The models were assessed using three prompt types: zero shot, guidelines enhanced, and few shot, and their performance was measured in terms of accuracy, precision, recall, F1 score, and processing speed. Results indicate that Claude 3.5 Sonnet consistently outperformed the other models across all metrics, demonstrating superior comprehension and classification abilities. Error analysis revealed specific areas for improvement, particularly in logical reasoning and adherence to guidelines. The findings highlight the potential of LLMs to enhance diagnostic processes and reduce healthcare costs, while also emphasizing the need for further research to address data privacy, interoperability, and ethical considerations for clinical integration. Future efforts should focus on leveraging open-source models and expanding datasets to optimize LLMs for real-world medical applications.**

*Keywords— Chain of Thought, Claude 3.5 Sonnet, Clinical Rationale Generation, Large Language Models, Medical Guidelines Prompt*

## I. Introduction

Pulmonary function tests (PFTs) are essential diagnostic tools used to assess lung health and diagnose various pulmonary conditions, including chronic obstructive pulmonary disease (COPD), asthma, and restrictive lung diseases [1-3]. The Global Initiative for Chronic Obstructive Lung Disease (GOLD) report highlights that COPD is a leading cause of morbidity and mortality worldwide, necessitating accurate diagnostic methods [4]. Traditionally, PFTs are conducted using laboratory-based spirometers, which provide precise measurements but come with high costs and require skilled technicians. This makes the approach impractical and leads to substantial increases in medical expenses for patients directed to tertiary hospitals [5].

Several studies have indicated that measurements obtained using portable spirometers are nearly identical to those from traditional, laboratory-based spirometers [6, 7]. This suggests that portable spirometers, with their cost-effectiveness, portability, and ease of use, can serve as a viable alternative to conventional PFTs devices [8]. Recently, the integration of portable spirometry with mobile applications has enabled patients to easily measure their lung capacity at home and transmit the results to healthcare providers. This advancement goes beyond the development of remote treatment plans, extending to the creation of diverse software services related to pulmonary health [9-11].

Moreover, with the advent of large language models (LLMs), the medical field has seen substantial advancements [12], including the development of medical domain LLMs capable of passing medical licensing exams [13], as well as applications in disease prediction and diagnostic consultation chatbots [14]. Ongoing research is also aimed at creating LLMs that can perform tasks beyond simple algorithmic classification of pulmonary function test data, generating pulmonary function test charts at the proficiency level of medical residents [15]. Such innovations have the potential to significantly reduce unnecessary medical expenditures and alleviate the burden of frequent hospital visits for patients.

This research was supported by the Bio&Medical Technology Development Program of the National Research Foundation (NRF) funded by the Korean government (MSIT) (No. RS-2024-00440371) and by the NRF grant funded by the Korean government (MSIT) (No. RS-2024-00457381).

Therefore, the aim of this study is to conduct foundational research by evaluating and comparing the comprehension of pulmonary function test data among three leading state-of-the-art large parameter LLMs, namely the Gemini 1.5 Pro [16], the GPT 4o [17] and the Claude 3.5 Sonnet [18], which have demonstrated top tier performance across various assessments.

## II. METHODOLOGY

The methodology section provides a comprehensive overview of the experimental design, data collection, preprocessing, and the specific approaches used to evaluate the performance of three advanced LLMs in interpreting PFTs data.

### A. Data Collection and Preprocessing

The study involved a total of 200 participants who were recruited from various primary healthcare settings. Participants included individuals across different age groups, genders, and varying health statuses to ensure a representative sample. Each participant provided informed consent before participating in the study, ensuring ethical standards were maintained.

Data were collected using the portable spirometer BULO M, developed by BREATHINGS Co., Ltd. This device is known for its accuracy and portability, which makes it ideal for use in both clinical and home settings. The spirometer measured two key pulmonary parameters: forced vital capacity (FVC) and forced expiratory volume in one second (FEV1). These parameters are critical in diagnosis of pulmonary diseases.

The predicted normal values for FVC and FEV1, used to classify pulmonary disease types, were calculated using Dr. Choi's equation [19], a widely accepted formula for predicting normal values based on the Asian population and considering age, sex, height, and weight.

This normalization was essential to account for individual variations and ensure that the comparisons made across participants were valid.

Using the predicted normal values for FVC and FEV1, the data were classified into four pulmonary disease categories: normal, restrictive, obstructive, or combined. The classification followed established diagnostic guidelines:

- Normal: FEV1/FVC ratio of 70% or higher and FVC 80% or higher of the predicted normal value for FVC; labeled as 'normal'.

- Restrictive: FEV1/FVC ratio of 70% or higher and FVC less than 80% of the predicted normal value for FVC; labeled as 'restrictive'.

- Obstructive: FEV1/FVC ratio of less than 70% and FVC 80% or higher of the predicted normal value for FVC; labeled as 'obstructive'.

- Combined: Meeting the criteria for both restrictive and obstructive conditions; labeled as 'combined'.

The ground truth labels for each participant were determined based on the above contents.

To facilitate the evaluation of the LLMs, the numerical PFTs data were transformed into standardized sentences. These sentences followed a specific structure, as outlined in the study's guidelines, to ensure consistency and clarity. Each sentence included the measured FVC and FEV1 values and the predicted normal values for these measurements to maintain uniformity.

### B. Large Language Models

The study evaluated three state of the art LLMs: Gemini 1.5 Pro (developed by Google), GPT 4o (developed by OpenAI), and Claude 3.5 Sonnet (developed by Anthropic). These models were chosen based on their superior performance across various natural language processing tasks and their availability via application programming interfaces.

Each model was accessed through its respective application programming interface and configured with specific parameters to ensure consistent performance across all tests:

- Temperature: Set to 0.0 to ensure deterministic outputs. This parameter controls the randomness of the model's predictions, with a lower temperature resulting in more consistent and predictable responses.

- Top p: Set to 1.0, which means the entire probability distribution of possible tokens was considered during the generation of responses. This setting ensures that no potential answers are excluded from consideration.

### C. Prompts for Pulmonary Disease Type Classification

Prompt engineering is a crucial aspect of evaluating LLMs, as the quality and structure of prompts can significantly impact the models' performances. This study employed three types of prompts to assess the models' abilities to interpret PFTs data and generate accurate clinical rationales.

Each case was processed in a separate session to ensure that prior inputs did not influence subsequent responses, maintaining the integrity and independence of each test case.

Zero shot prompting involves providing the model with PFTs data without any additional context or examples. This type of prompt assesses the model's inherent ability to understand and interpret the data based solely on its pre-existing knowledge. Each model was given a prompt structured as follows, corresponding to Fig. 1:

- Entry of transformed PFTs data.

- A request to generate a diagnosis (normal, restrictive, obstructive, or combined).

- A request to provide a clinical rationale in four sentences.

To improve the models' performance, zero shot prompt were enhanced with specific diagnostic guidelines and the chain of thought (CoT) technique [20]. The guidelines provided explicit criteria for classifying pulmonary diseases, while the CoT technique involved breaking down the reasoning process into sequential steps. The guidelines enhanced prompt structure included the following, corresponding to Fig. 1:

- The same PFTs data as provided in the zero shot prompt.

- Detailed diagnostic guidelines to assist the model in making accurate classifications.

- CoT sentences to guide the model through the reasoning process step by step.

**Zero Shot Prompt**

As a pulmonologist, referencing the patient's **[Pulmonary Function Test Results]**, diagnose one of the four types of pulmonary disease: 'normal', 'restrictive', 'obstructive', or 'combined', and generate clinical rationale for the diagnosis.

**[Pulmonary Function Test Results]**: FVC (Forced Vital Capacity): {} L, FEV1 (Forced Expiratory Volume in one second): {} L, Predicted normal value for FVC: {} L, Predicted normal value for FEV1: {} L

**Diagnosis**: 'normal', 'restrictive', 'obstructive', or 'combined' **Clinical Rationale**: Please express this in four sentences

**Guidelines Enhanced Prompt**

**[Diagnostic Guidelines]** :
1. If the FEV1/FVC ratio is 70% or greater, and the FVC value is 80% or more of the predicted normal value for FVC, classify as 'normal'.
2. If the FEV1/FVC ratio is 70% or greater, and the FVC value is less than 80% of the predicted normal value for FVC, classify as 'restrictive'.
3. If the FEV1/FVC ratio is less than 70%, and the FVC value is 80% or more of the predicted normal value for FVC, classify as 'obstructive'.
4. If the FEV1/FVC ratio is less than 70%, and the FVC value is less than 80% of the predicted normal value for FVC, classify as 'combined'.

**[Pulmonary Function Test Results]**: FVC (Forced Vital Capacity): {} L, FEV1 (Forced Expiratory Volume in one second): {} L, Predicted normal value for FVC: {} L, Predicted normal value for FEV1: {} L

Apply the **[Diagnostic Guidelines]** to the **[Pulmonary Function Test Results]** in order, following steps 1, 2, 3, and 4, and make a diagnosis.

**Diagnosis**: 'normal', 'restrictive', 'obstructive', or 'combined' **Clinical Rationale**: Please express this in four sentences.

**Few Shot Prompt**

**[Example 1]** : **[Pulmonary Function Test Results]**: FVC (Forced Vital Capacity): 5.15 L, FEV1 (Forced Expiratory Volume in one second): 4.09 L, Predicted normal value for FVC: 4.98 L, Predicted normal value for FEV1: 4.15 L
**Diagnosis**: 'normal', **Clinical Rationale**: This patient's FEV1/FVC ratio is approximately 79.42%. This is greater than 70%, indicating a potential 'normal' or 'restrictive' type. The FVC/Predicted normal value for FVC ratio is approximately 103.41%. This is greater than 80%, so the diagnosis for this patient is 'normal'.

**[Example 2]** : **[Pulmonary Function Test Results]**: FVC (Forced Vital Capacity): 3.48 L, FEV1 (Forced Expiratory Volume in one second): 3.18 L, Predicted normal value for FVC: 4.52 L, Predicted normal value for FEV1: 3.96 L
**Diagnosis**: 'restrictive', **Clinical Rationale**: This patient's FEV1/FVC ratio is approximately 91.38%. This is greater than 70%, indicating a potential 'normal' or 'restrictive' type. The FVC/Predicted normal value for FVC ratio is approximately 76.99%. This is less than 80%, so the diagnosis for this patient is 'restrictive'.

**[Example 3]** : **[Pulmonary Function Test Results]**: FVC (Forced Vital Capacity): 4.37 L, FEV1 (Forced Expiratory Volume in one second): 2.56 L, Predicted normal value for FVC: 4.16 L, Predicted normal value for FEV1: 3.45 L
**Diagnosis**: 'obstructive', **Clinical Rationale**: This patient's FEV1/FVC ratio is approximately 58.58%. This is less than 70%, indicating a potential 'obstructive' or 'combined' type. The FVC/Predicted normal value for FVC ratio is approximately 105.05%. This is greater than 80%, so the diagnosis for this patient is 'obstructive'.

**[Example 4]** : **[Pulmonary Function Test Results]**: FVC (Forced Vital Capacity): 2.89 L, FEV1 (Forced Expiratory Volume in one second): 1.83 L, Predicted normal value for FVC: 4.39 L, Predicted normal value for FEV1: 3.89 L
**Diagnosis**: 'combined', **Clinical Rationale**: This patient's FEV1/FVC ratio is approximately 63.32%. This is less than 70%, indicating a potential 'obstructive' or 'combined' type. The FVC/Predicted normal value for FVC ratio is approximately 65.83%. This is less than 80%, so the diagnosis for this patient is 'combined'.

Fig. 1. Prompts for classification of pulmonary disease types and generation of clinical rationale

Few shot prompting involves providing the model with a small number of example cases in addition to the primary prompt. This approach helps the model understand the task better by learning from the examples. For this study, four example datasets were included in the prompts. The structure of the few shot prompt, as shown in Fig. 1, was as follows:

- The same PFTs data input as the previous two prompts.
- The same diagnostic guidelines and CoT sentences input as the previous prompts.
- Input of four example responses.

### D. Evaluation for Classification

The quantitative performance of the LLMs in classifying different types of pulmonary diseases was assessed through a variety of evaluation metrics:

- Accuracy: The proportion of correct predictions out of the total predictions.
- Precision: The ratio of true positives to the sum of true positives and false positives, indicating the model's ability to avoid false positives and its reliability in positive predictions.

- Recall: The ratio of true positives to the sum of true positives and false negatives, reflecting the model's ability to identify all relevant instances and its effectiveness in capturing actual positives.

- F1 Score: The harmonic mean of precision and recall, providing a balanced measure of the model's performance, particularly useful when there is an uneven class distribution.

These metrics were selected to ensure a comprehensive understanding of how well the models performed in distinguishing between the four categories of pulmonary conditions.

### E. Error Types

The clinical rationales generated by the LLMs were analyzed for different types of errors, including:

- Comparison Operation Errors (COE): These errors occurred when the model's logical operations involving numerical comparisons were flawed, despite the calculations being correct.

- Guideline Deviations (GD): These errors happened when the model deviated from the provided diagnostic

guidelines, either by offering alternative explanations or additional clinical opinions that were not aligned with the guidelines.

- Misdiagnoses (MD): In these cases, the model generated a clinical rationale that aligned with the diagnostic guidelines, but the final diagnosis was incorrect.

- Miscalculations (MC): These errors involved incorrect calculations during the reasoning process, which could lead to incorrect conclusions.

## III. RESULTS

In this section, we present the detailed results of our evaluation of the three LLMs: Gemini 1.5 Pro, GPT 4o, and Claude 3.5 Sonnet, on their comprehension of PFTs data. The evaluation was conducted across three types of prompts: zero shot, guidelines enhanced, and few shot. This includes the quantitative performance metrics, such as accuracy, precision, recall, and F1 score, along with the time efficiency and the number of tokens generated by the three models for 200 iterations to assess their processing speed and efficiency under different prompt types. Additionally, we present a comprehensive error analysis to provide insights into the models' capabilities and limitations.

### A. Quantitative Performance and Time Efficiency of Pulmonary Disease Type Classification by Prompt

The zero shot performance of the models was evaluated by providing them with only the PFTs results without any additional context. Table I summarizes the accuracy, precision, recall, and F1 score for each model across the four pulmonary disease types: normal, restrictive, obstructive, and combined.

Claude 3.5 Sonnet outperformed the other models in the zero shot setting across all disease types, achieving the highest accuracy, precision, recall, and F1 score. This indicates that Claude 3.5 Sonnet has a superior inherent ability to understand and classify pulmonary diseases based solely on the provided PFTs data.

TABLE I. PERFORMANCE EACH MODEL IN CLASSIFYING PULMONARY DISEASE TYPES WITH ZERO SHOT PROMPT

| Model | Pulmonary Disease type | Precision | Recall | F1 Score | Overall Accuracy |
|---|---|---|---|---|---|
| Gemini 1.5 Pro | Normal | 1.00 | 0.13 | 0.24 | |
| | Restrictive | 0.81 | 0.66 | 0.72 | 0.55 |
| | Obstructive | 0.43 | 1.00 | 0.60 | |
| | Combined | 0.00 | 0.00 | 0.00 | |
| GPT 4o | Normal | 1.00 | 0.48 | 0.65 | |
| | Restrictive | 0.81 | 0.92 | 0.86 | 0.73 |
| | Obstructive | 0.60 | 0.98 | 0.75 | |
| | Combined | 0.00 | 0.00 | 0.00 | |
| Claude 3.5 Sonnet | Normal | 0.92 | 0.77 | 0.84 | |
| | Restrictive | 0.71 | 1.00 | 0.83 | 0.80 |
| | Obstructive | 1.00 | 0.72 | 0.83 | |
| | Combined | 0.41 | 0.44 | 0.42 | |

The guidelines enhanced prompt included specific diagnostic criteria and the CoT technique to aid the models in their reasoning process. Table II presents the performance metrics for each model using the guidelines enhanced prompt.

With the inclusion of diagnostic guidelines and CoT, overall accuracy showed minimal change, but there was an improvement in the classification of the 'combined' type. The Claude 3.5 Sonnet maintained its lead, achieving the highest scores across all metrics and disease types. The enhancements provided clearer criteria and reasoning steps, which helped the models make more accurate diagnoses.

The few shot prompt included four example datasets to provide additional context and improve the models' understanding. Table III details the performance metrics for each model with the few shot prompt.

The few shot prompting further improved the performance of all models, with Claude 3.5 Sonnet continuing to outperform Gemini 1.5 Pro and GPT 4o. The additional examples provided context that helped the models better understand the classification task, leading to higher accuracy and more reliable predictions.

TABLE II. PERFORMANCE EACH MODEL IN CLASSIFYING PULMONARY DISEASE TYPES WITH GUIDELINES ENHANCED PROMPT

| Model | Pulmonary Disease type | Precision | Recall | F1 Score | Overall Accuracy |
|---|---|---|---|---|---|
| Gemini 1.5 Pro | Normal | 1.00 | 0.22 | 0.36 | |
| | Restrictive | 0.47 | 0.92 | 0.62 | 0.48 |
| | Obstructive | 0.74 | 0.28 | 0.41 | |
| | Combined | 0.21 | 0.50 | 0.30 | |
| GPT 4o | Normal | 0.88 | 0.50 | 0.64 | |
| | Restrictive | 0.62 | 0.95 | 0.75 | 0.73 |
| | Obstructive | 0.79 | 0.83 | 0.81 | |
| | Combined | 1.00 | 0.31 | 0.48 | |
| Claude 3.5 Sonnet | Normal | 0.92 | 0.57 | 0.70 | |
| | Restrictive | 0.69 | 0.97 | 0.81 | 0.81 |
| | Obstructive | 0.98 | 0.83 | 0.90 | |
| | Combined | 0.73 | 1.00 | 0.84 | |

TABLE III. PERFORMANCE EACH MODEL IN CLASSIFYING PULMONARY DISEASE TYPES WITH FEW SHOT PROMPT

| Model | Pulmonary Disease type | Precision | Recall | F1 Score | Overall Accuracy |
|---|---|---|---|---|---|
| Gemini 1.5 Pro | Normal | 0.97 | 0.50 | 0.66 | |
| | Restrictive | 0.74 | 0.70 | 0.72 | 0.71 |
| | Obstructive | 0.75 | 0.88 | 0.81 | |
| | Combined | 0.41 | 0.94 | 0.57 | |
| GPT 4o | Normal | 1.00 | 0.47 | 0.64 | |
| | Restrictive | 0.68 | 0.98 | 0.80 | 0.81 |
| | Obstructive | 0.93 | 0.95 | 0.94 | |
| | Combined | 0.83 | 0.94 | 0.88 | |
| Claude 3.5 Sonnet | Normal | 0.97 | 0.65 | 0.78 | |
| | Restrictive | 0.73 | 1.00 | 0.84 | 0.84 |
| | Obstructive | 0.93 | 0.85 | 0.89 | |
| | Combined | 0.88 | 0.94 | 0.91 | |

## B. Time Efficiency of Three Models by Prompt Type

Table IV presents the time required for three LLMs, Gemini 1.5 Pro, GPT 4o, and Claude 3.5 Sonnet across different prompt types: zero shot, guidelines enhanced, and few shot.

TABLE IV. TIME EFFICIENCY OF THREE MODELS ACROSS DIFFERENT PROMPT TYPES

| Model | Prompt Type | Total Time (seconds) | Average Time per Iteration (seconds) | Average Tokens per Iteration (tokens) |
|---|---|---|---|---|
| Gemini 1.5 Pro | Zero Shot | 531.94 | 2.66 | 63.99 |
| | Guidelines Enhanced | 688.14 | 3.44 | 77.84 |
| | Few Shot | 676.78 | 3.38 | 70.19 |
| GPT 4o | Zero Shot | 489.86 | 2.45 | 88.93 |
| | Guidelines Enhanced | 557.63 | 2.79 | 100.00 |
| | Few Shot | 348.57 | 1.74 | 47.33 |
| Claude 3.5 Sonnet | Zero Shot | 751.22 | 3.75 | 92.03 |
| | Guidelines Enhanced | 967.45 | 4.84 | 99.71 |
| | Few Shot | 1090.10 | 5.04 | 105.44 |

Gemini 1.5 Pro showed intermediate performance, with average times per iteration ranging from 2.66 to 3.44 seconds, depending on the prompt type. For the zero-shot prompt, GPT 4o had the shortest total time (489.86 seconds) and average time per iteration (2.45 seconds), with 88.93 tokens per iteration.

In contrast, Claude 3.5 Sonnet took the most time for all prompt types, particularly with the few shot prompt, where the total time was 1090.10 seconds and the average per iteration was 5.04 seconds.

## C. Error Rate

In this study, we conducted a detailed error analysis to understand the types and frequencies of errors made by Gemini 1.5 Pro, GPT 4o, and Claude 3.5 Sonnet in interpreting PFTs data.

Fig. 2 illustrates the error rate for each model across the different prompt types (zero shot, guidelines enhanced, and few shot). The results reveal that Claude 3.5 Sonnet consistently outperformed both Gemini 1.5 Pro and GPT 4o in all prompt settings, achieving the lowest error rate.

In the zero shot setting, Claude 3.5 Sonnet achieved an error rate of 20%, significantly lower than GPT 4o (26%) and Gemini 1.5 Pro (45%). This indicates that Claude 3.5 Sonnet has a superior inherent ability to understand and classify pulmonary disease types based on PFTs data without additional context.

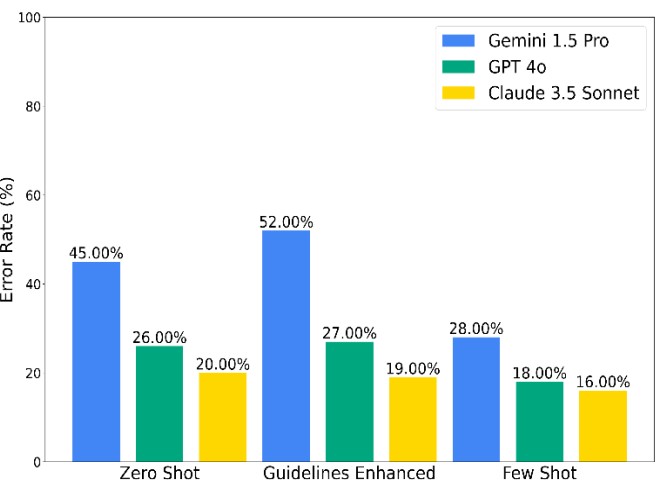

Fig. 2. Error rate in the classification of pulmonary disease types

The introduction of diagnostic guidelines and the CoT technique resulted in improved performance for all models. Claude 3.5 Sonnet again demonstrated the lowest error rate at 19%, followed by GPT 4o at 27% and Gemini 1.5 Pro at 52%. Despite the addition of guidelines, the error rate for both Gemini 1.5 Pro and GPT 4o slightly increased.

The inclusion of four example datasets further reduced the error rate across all models. Claude 3.5 Sonnet achieved the lowest error rate at 16%, GPT 4o improved to 18%, and Gemini 1.5 Pro showed a notable reduction to 28%. The few shot learning approach significantly enhanced the models' ability to leverage additional context, resulting in more accurate classifications.

## D. Types of Errors in Few Shot Prompt

Fig. 3 categorizes the examples of errors generated by the LLMs into four types. COE referred to instances where the calculations were correct, but the logical operation involving comparisons with values such as 70 or 80 was flawed. GD indicated cases where the LLMs deviated from the provided diagnostic guidelines during the second and third prompts, providing alternative explanations or additional clinical opinions.

MD occurred when the clinical rationale aligns with the diagnostic guidelines, but the "diagnosis:" field contained an incorrect answer. Lastly, MC denoted calculation errors, which were observed only in the Gemini 1.5 Pro model. These errors cooccurred with 3 instances of MD, 6 instances of GD, and 10 instances of COE.

Fig. 4 illustrates the number of errors by type generated by each model for the few shot prompt. The Gemini 1.5 Pro model displayed a balanced distribution of errors, including COE, GD, and MD, as well as calculation mistakes. The GPT 4o model exhibited relatively fewer errors overall, most of which were of the COE type. The Claude 3.5 Sonnet model had the fewest errors overall, with the most common type being GD, indicating its tendency to provide additional clinical insights.

**[Examples of Errors]**

**[Comparison Operation Error, COE]:**

" ~ (omission) The FVC ratio is approximately 82.70%. This is less than 80%."

**[Guidelines Deviation, GD]:**

" ~ (omission) Given that the FEV1/FVC ratio is very close to the 70% threshold and the FVC is well-preserved, a 'restrictive' diagnosis is most appropriate, (omission) ~."

**[MisDiagnosis, MD]:**

"Diagnosis: 'restrictive'"
"Clinical Rationale: ~ (omission) the diagnosis is 'obstructive'."

**[MisCalculation, MC]:**
"~ (omission) This patients's FEV1/FVC ratio is approximately 89.71% (2.8/3.11), (omission) ~."
Actual : 90.03% (2.8/3.11)

Fig. 3. Examples of errors generated by models in few shot prompt

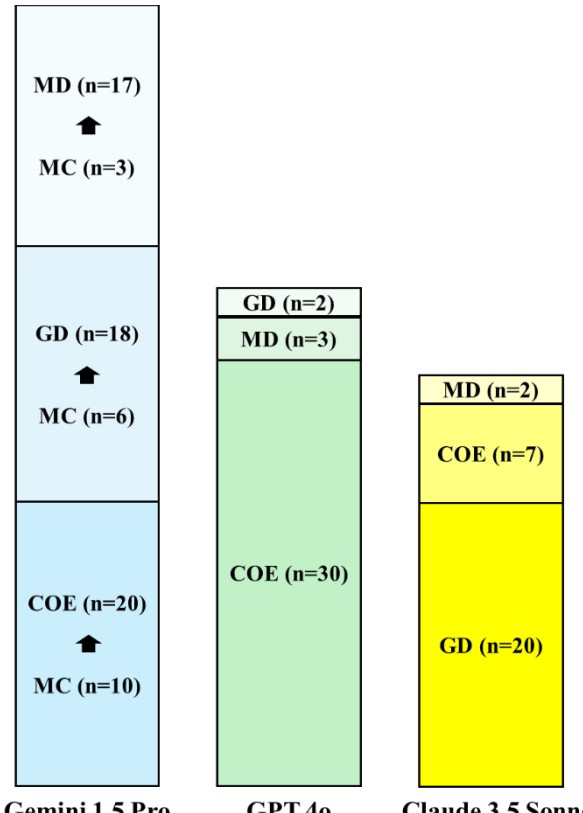

Fig. 4. Number of errors generated by each model in few shot prompt

The Gemini 1.5 Pro displayed a more balanced distribution of error types, including significant instances of COEs, GDs, and MDs. This indicates that while the model had some understanding of the guidelines, it frequently made logical errors and provided incorrect diagnoses despite correct rationales, compounded further by instances of MCs.

The GPT 4o, with relatively fewer errors, showed a predominant issue with COEs, suggesting that the model's main challenge was in the logical interpretation of numerical data rather than following the guidelines or providing accurate diagnoses.

The Claude 3.5 Sonnet generated the least number of errors overall, with the majority being GDs. The tendency of Claude 3.5 Sonnet to deviate slightly from the guidelines while still offering clinically relevant insights indicates its advanced understanding of the context, albeit with a propensity to provide more information than necessary.

## IV. DISCUSSION

The results highlight the potential of advanced LLMs, particularly the Claude 3.5 Sonnet, in the medical field for diagnosing pulmonary diseases. Although Claude 3 Opus is a different model from the Claude 3.5 Sonnet and is not specialized for the medical domain, it demonstrated performance [21] on the PubMedQA dataset [22] nearly equivalent to Google's Med-PaLM 2 [23], which is designed specifically for medical applications. This suggests that the contexts of our experiment and those results are similar.

The ability of these models to accurately interpret PFTs data and generate clinical rationales can significantly enhance diagnostic processes [24], reduce healthcare costs [25], and improve patient outcomes [26, 27]. Integrating such AI technologies into clinical settings can provide healthcare professionals with reliable decision support tools [28], leading to more efficient and accurate diagnoses [29], and potentially advancing to specific tasks such as summarizing risk levels based on preoperative pulmonary function test results [30].

However, implementing LLMs in clinical settings requires careful consideration of data privacy and integration with existing systems [31]. Key steps include ensuring data security through local deployment and anonymization, as well as complying with healthcare regulations like the Health Insurance Portability and Accountability Act (HIPAA) and establishing interoperability with electronic health record (EHR) systems using a standardized application programming interface (API) [32, 33]. Additionally, fine tuning LLMs with domain specific data and incorporating human oversight are crucial for enhancing accuracy and maintaining clinical standards [34]. Therefore, balancing technological advancements with ethical and practical considerations will be critical for the successful integration of LLMs in healthcare [35].

Furthermore, for the future development of LLMs capable of generating pulmonary function test charts for use in real world medical settings, it is essential to utilize open sourced LLMs [36]. Related research has demonstrated the application of large parameter LLMs, such as GPT 4, to clinical data for diagnosing Alzheimer's disease and generating clinical

rationales, subsequently using knowledge distillation to train open sourced models [37].

The key aspect is that large parameter LLMs generate instruction following data for training open sourced models [38]. This study's significance lies in comparing three state of the art models, which currently demonstrate superior performance, using PFTs data, and analyzing their errors to provide insights into the understanding of pulmonary diseases by large parameter LLMs.

Additionally, exploring and fine tuning other open sourced LLMs can help identify models with even greater potential for clinical applications [39]. Further efforts should focus on minimizing specific types of errors, such as COEs and GDs, to enhance the models' reliability and accuracy [40].

## V. CONCLUSION

This study evaluated the performance of three state of the art LLMs: Gemini 1.5 Pro, GPT 4o, and Claude 3.5 Sonnet in interpreting PFTs data and generating clinical rationales for diagnosing pulmonary diseases. Notably, there are no recent studies that directly compare the performance of these latest models on specific clinical tasks, making this research a novel contribution to the field.

The Claude 3.5 Sonnet consistently outperformed the Gemini 1.5 Pro and the GPT 4o across all evaluation metrics and prompt types. It demonstrated the highest accuracy, precision, recall, and F1 score, indicating its superior ability to understand and classify PFTs data. The model's performance was enhanced by the inclusion of diagnostic guidelines and few shot examples, showcasing its capacity to learn effectively from additional context. The Gemini 1.5 Pro and the GPT 4o also showed improvements with enhanced prompts, but their overall performance remained below that of the Claude 3.5 Sonnet.

From the perspective of error analysis, the Gemini 1.5 Pro exhibited a balanced distribution of errors, including COEs, GDs, and MDs, indicating a need for better logical reasoning and adherence to guidelines. The GPT 4o, while demonstrating fewer errors overall, struggled primarily with COEs, suggesting a need for improved numerical interpretation capabilities.

While this study provides valuable insights, it is limited by the sample size and the specific models evaluated. Future research should involve larger and more diverse datasets to better generalize the findings. Additionally, further efforts should focus on minimizing specific types of errors, such as COEs and GDs, to enhance the models' reliability and accuracy.

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
