# OpenReview forum: "Evaluating Advanced Large Language Models for Pulmonary Disease Diagnosis Using Portable Spirometer Data: A Comparative Analysis of Gemini-1.5 Pro, GPT-4o, and Claude-3.5 Sonnet"
_IEEE.org/EMBS/BHI/2024/Conference — IEEE BHI'24_

### Official Review · Reviewer_P4ao · 2024-07-31
**Evaluating Advanced Large Language Models for Pulmonary Disease Diagnosis Using Portable Spirometer Data: A Comparative Analysis of Gemini 1.5 Pro, GPT 4o , and Claude 3.5 Sonnet**

**Overall Rating:** 5
**Confidence:** 3

**Other Quality Metrics:**

•	Some more references need to be added.

**Questions For The Authors:**

•	Tables I, II, III in the captions don’t match with Tables 1, 2, 3 in the discussion.

**Strengths:**

•	Results indicate that Claude-3.5 Sonnet consistently outperformed the other models.

**Summary Of The Paper:**

•	The study aims to evaluate and compare the performance of three advanced LLMs—Gemini-1.5 Pro, GPT-4o, and Claude-3.5 Sonnet—in understanding and interpreting PFTs data.

**Weaknesses:**

•	Relevant references are less.

---

### Official Review · Reviewer_Liuo · 2024-08-01
**Good and Interesting Paper**

**Overall Rating:** 7
**Confidence:** 4

**Other Quality Metrics:**

(a) Clarity of writing - Good
(b) Clinical Significance - Good
(c) Methodological Novelty - Great
(d) Experiments and Results - Good

**Questions For The Authors:**

1.	How representative are these cases of the broader population? Is it possible for demographic differences to impact the performance of such models?
2.	What specific steps would be necessary to implement these LLMs in a clinical setting, particularly concerning data privacy and integration with existing medical systems?

**Strengths:**

1.	The paper effectively presents the reasoning behind the need for exploitation of LLM for pulmonary disease diagnosis. This provides a justification of the importance and the impact of the study.
2.	The paper appropriately discusses the limitations of the study and outline the future steps.
3.	This work is interesting and sounds promising since LLM are currently very popular. As it can be seen from the results of the paper, LLM can be beneficial in the medical field for the diagnosis of pulmonary diseases, with Claude-3.5 Sonnet being the most accurate model.

**Summary Of The Paper:**

The paper evaluated the performance of three leading state-of-the-art Advanced Large Language Models (LLM), Gemini-1.5 Pro, GPT-4o, and Claude-3.5 Sonnet, for pulmonary disease diagnosis using data from portable spirometers. The study included 200 participants and the models were tested through zero-shot, guidelines-enhanced, and few-shot prompts. The numerical data were transformed into standardized sentences following a specific structure for consistency and clarity. Performance metrics included accuracy, precision, recall, and F1-score, with Claude-3.5 Sonnet outperforming the other models.

**Weaknesses:**

1.	Relatively small dataset. Additional data must be included to ensure the generalizability of the models.
2.	The paper does not report the time required for the LLMs to evaluate the data. Time metrics can be beneficial in order to assess the efficiency of the models in case of time-sensitive decision-making.

Formatting issues.
- The margins on the first page are not correctly set according to the rest of the paper.
- The paper does not enable hyphenation.
- Section II-E is not formatted in a column-wise manner. Please check the entire manuscript for this.
- Tables are very large, with large margins and font size.

---

### Official Review · Reviewer_MJvR · 2024-08-12
**The paper considers large language models for pulmonary disease diagnosis. It is well-written paper and covers a versatile analysis.**

**Overall Rating:** 8
**Confidence:** 3

**Other Quality Metrics:**

(a) Clarity of writing: Great
(b) Clinical significance: Great
(c) Methodological novelty:Good
(d) Experiments and Results:Good

**Questions For The Authors:**

I have no questions for the authors.

**Strengths:**

The problem considered would improve the life quality of patients. The paper is well-written and the results are given expilictly with tables. All the steps of anaysis is explained and the results are disscussed throughly.

**Summary Of The Paper:**

In the paper pulmonary function tests (PFTs) which are crucial for the accurate diagnosis and prognosis of pulmonary diseases are considered. Large language models (LLMs) are considered to overcome the diffuculties of traditional spirometers. With the software provided, the integration of portable spirometry with mobile applications has enabled patients to easily measure their lung capacity at home and transmit the results to healthcare providers. For this application three different LLMs, Gemini-1.5 Pro, GPT-4o, and Claude-3.5 Sonnet, are considered to understand and interpret PFTs data. Data from 200 participants are used and the models are tested through zero-shot, guidelines-enhanced, and few-shot prompts. Performance metrics included accuracy, precision, recall, and F1-score for disease classification, along with error analysis of clinical rationale. Results indicate that Claude-3.5 Sonnet consistently outperformed the other models.

**Weaknesses:**

The words at the end of the lines are cut improperly, making it difficult to follow, please fix this small problem.

---

### Decision · Program_Chairs · 2024-09-23

Accept